# A Sporadic and Lethal Lassa Fever Case in Forest Guinea, 2019

**DOI:** 10.3390/v12101062

**Published:** 2020-09-23

**Authors:** N’Faly Magassouba, Enogo Koivogui, Sory Conde, Moussa Kone, Michel Koropogui, Barrè Soropogui, Ifono Kekoura, Julia Hinzmann, Stephan Günther, Sakoba Keita, Sophie Duraffour, Elisabeth Fichet-Calvet

**Affiliations:** 1Laboratoire des Fièvres Hémorragiques en Guinée, Conakry, Guinea; cmagassouba01@gmail.com (N.M.); koropom@gmail.com (M.K.); barresoropogui@gmail.com (B.S.); ifonokekoura@gmail.com (I.K.); 2Agence Nationale de Sécurité Sanitaire, Ministry of Health, Conakry, Guinea; enogo78@gmail.com (E.K.); soryconde25@gmail.com (S.C.); moisek@gmail.com (M.K.); sakoba54@gmail.com (S.K.); 3Virology Department, Bernhard Nocht Institute for Tropical Medicine, 20359 Hamburg, Germany; hinzmann@bnitm.de (J.H.); guenther@bnitm.de (S.G.); duraffour@bnitm.de (S.D.); 4Virology Department, German Center for Infection Research (DZIF), partner site Hamburg–Lübeck–Borstel–Riems, 20359 Hamburg, Germany

**Keywords:** Lassa fever, Kissidougou, Guinea, Lassa genome, phylogeny

## Abstract

Lassa fever is a rodent-borne disease caused by Lassa virus (LASV). It causes fever, dizziness, vertigo, fatigue, coughing, diarrhea, internal bleeding and facial edema. The disease has been known in Guinea since 1960 but only anectodical acute cases have been reported to date. In January 2019, a 35-year-old man, a wood merchant from Kissidougou, Forest Guinea, presented himself at several health centers with persistent fever, frequent vomiting and joint pain. He was repeatedly treated for severe malaria, and died three weeks later in Mamou regional hospital. Differential diagnosis identified LASV as the cause of death. No secondary cases were reported. The complete LASV genome was obtained using next-generation sequencing. Phylogenetic analysis showed that this strain, namely the Kissidougou strain, belongs to the clade IV circulating in Guinea and Sierra Leone, and is thought to have emerged some 150 years ago. Due to the similarity of symptoms with malaria, Lassa fever is still a disease that is difficult to recognize and that may remain undiagnosed in health centers in Guinea.

## 1. Introduction

Lassa fever is a haemorrhagic fever due to Lassa virus (LASV) that was discovered in 1969 in Nigeria [1,2]. The disease is endemic in West Africa, particularly in Guinea, Sierra Leone, Liberia, Mali, Côte d’Ivoire, Togo, Benin and Nigeria. The majority of cases are registered in Sierra Leone and Nigeria, while the other countries have registered sporadic cases [3,4,5]. In Guinea, the disease is widespread in the south, as evidenced by seroprevalence rates in humans ranging from 20% to 40% in contrast to the 3% to 8% observed in the north [6]. Ecological studies on rodent reservoirs show a similar distribution of the Lassa virus [7,8,9,10,11]. 

However, Guinea has very few reported acute human cases, and only a few studies have a posteriori described acute Lassa fever cases in the prefectures of Kindia, Faranah, Kissidougou, Guekedou, Macenta and N’Zérékoré in 1992 and 1996–1999 [12,13,14]. Therefore, the emergence of one acute case in 2018 in Macenta [15] and another one in 2019 in Kissidougou [16] has prompted great interest, necessitating immediate investigation. The epidemic potential of Lassa fever was recently revised by the Word Health Organization, which has listed Lassa fever among the top five zoonoses to be monitored, along with Ebola, MERS-CoV, Nipah and Zika (https://www.who.int/teams/blueprint). In this article, we describe both the epidemiological and molecular investigation of the Kissidougou case and his contacts.

## 2. Materials and Methods 

### 2.1. Field Investigation

Field investigations were managed by the Agence Nationale de Sécurité Sanitaire (ANSS) in consultation with members of the Ministry of Health, the Gamal-Nasser University of Conakry, the Word Health Organization (WHO), the NGO Médecins sans Frontières (MSF) and the Centre of Disease Control (CDC). A first investigation took place 6–7 February 2019 in the prefecture of Kissidougou where the patient lived. A second investigation took place 6–7 February 2019 in the Mamou prefecture where the patient died (Figure 1). Contacts in health centres and hospitals were identified and their blood was collected for PCR testing. The tests were carried out in the laboratory on 31 January 2019 for the patient and from 8 to 9 February 2019 for the contacts (Appendix A). All subjects and the family of the deceased patient gave their informed consent. The protocol has been approved by the Ethics Committee for Health Research in Guinea (permit n°129/CNERS/16, approval 11 October 2016).

### 2.2. Molecular and Serological Diagnosis 

The presence of antibodies against HIV-1/HIV-2 and yellow fever virus (YFV) was tested using the INSTI kit (Biolytical, Richmond, Canada) and ELISA (Pasteur Institute, Dakar, Senegal) respectively. The presence of the Hepatitis B virus (HBV) was tested using a rapid screening kit (CapitalBio Technology, Beijing, China). The presence of the Ebola virus (EBOV) and LASV was investigated by reverse transcription–polymerase chain reaction (RT-PCR). Viral RNA was extracted from serum samples using the QIAmp viral RNA kit (Qiagen, Hilden, Germany). In Conakry, the EBOV test was carried out by real-time RT-PCR, using the RealStar^®^ Filovirus Screen RT-PCR Kit 1.0 (altona Diagnostics, Hamburg, Germany) on a Smart Cycler II system (Cepheid, Sunnyvale, CA, USA). The LASV test was carried out using a conventional RT-PCR targeting the glycoprotein (GP) with the primers LVS36+ (5′-ACCGGGGATCCTAGGCATTT-3′) and LVS339- (5′-GTTCTTTGTGCAGGAMAGGGGCATKGTCAT-3′) [18]. The EBOV and LASV tests were performed on both the patient and contacts in Conakry (Appendix A). A confirmatory real-time RT-PCR LASV test using the RealStar^®^ Lassavirus RT-PCR kit 2.0 (altona Diagnostics) on the Rotor-Gene Q platform (Qiagen, Hilden, Germany) was performed on the serum sample of the patient in our satellite laboratory in Gueckedou. The LASV-specific RT-PCR kit targets both the S- and L-segments of LASV. 

### 2.3. LASV Sequencing

The serum sample from the patient was dried on a filter paper and sent to the Bernhard Nocht Institute for Tropical Medicine in Hamburg, Germany. Extraction and metagenomic library preparation for next generation sequencing (NGS) on the Illumina platform were performed as described previously [5]. Briefly, viral RNA was extracted directly from dried serum on filter paper using the QIAmp viral RNA kit (Qiagen), further digested with DNAse (TURBO DNase, Thermo Fisher Scientific, Carlsbad, USA), randomly reverse-transcribed and amplified using a Sequence-Independent Single-Primer Amplification (SISPA) approach. The Illumina sequencing library was prepared using the Nextera XT v2 Kit (Illumina, San Diego, USA) with 1 ng of SISPA-amplified cDNA, according to the manufacturer’s instructions, with a total of 12 cycles in the library amplification PCR, and further sequenced on a 2 × 300 bp Illumina MiSeq run. Majority consensus was obtained with bases called at a minimum depth of 20x and a support fraction of 70%. Any base location that did not fulfil the depth and support fraction was assigned an “N” IUPAC ambiguity notation. The complete sequences were submitted to GenBank under the names and accession numbers LASV/GUI/KIS-2019-L-segment #MT861993 and LASV/GUI/KIS-2019-S-segment #MT861994.

### 2.4. Phylogenetic Analysis

The nucleotide sequences of full-length glycoprotein precursor (GPC), nucleoprotein (NP) and polymerase (L) were aligned separately in three data sets including 41 sequences for each. The sequences were chosen to be representative of their clusters after a preliminary analysis done with 64 NP sequences only and elimination of similar or clone sequences. The alignment of nucleotides was realized according to the position of amino acids in the protein alignment. The phylogeny was inferred by the Bayesian Markov Chain Monte Carlo (MCMC) method implemented in the BEAST software, version 1.10.1 [19]. With the view to performing a time-calibrated phylogeny, the parameters were set in BEAUTI as follow: the tip dates at the year level, the substitution model as GTR + gamma and codon partition with positions 1, 2, 3, the clock model as strict or uncorrelated relaxed. A coalescent tree with a constant size population was set as prior. The length of the chain was 10 million, with echo states and log parameters every 10,000 steps. The xml files issued from BEAUTI were run in BEAST and checked in TRACER. After checking the effective sample size to be above 200 for all the parameters, the consensus trees were obtained in TreeAnnotator, and then visualized through FigTree (BEAST packages, https://beast.community/programs). 

## 3. Results

### 3.1. The Case Description

On 7 January 2019, a 35-year-old man consulted a clinic in Kissidougou because of fever accompanied by chills, dizziness, very frequent bilious vomiting and joint pains. Malaria was diagnosed and the patient was treated with paracetamol and quinine. Despite complying with the malaria treatment, symptoms remained unchanged for one week. He returned to the same clinic, which further referred him to the regional hospital located in Kissidougou on 21 January 2019. At the end of the consultation, a diagnosis of malaria and typhoid fever was made. He received a perfusion and a recommendation of hospitalization was made. However, the patient refused to be hospitalized and came back home. After three days of treatment, and feeling that there was no improvement, he went to another health centre. Three days later, he decided to travel to Mamou, where he stayed two days with his family. Noting that there was no improvement in his status, his parents decided to bring him to the regional hospital of Mamou on 28 January 2019, where he was admitted at 2 p.m. to the emergency room at the Epidemic Treatment Centre (CT-EPI). At the time of admission, the symptoms were as follows: fever and chills, dizziness, vomiting, joint pain, diarrhoea and prostration. A diagnosis of severe malaria was given and he received an intensive supportive antimalarial treatment (rehydration, artesunate, ampicillin, paracetamol, dicynone). At 3 p.m., the patient started bleeding with a cough tinged with blood and the epistaxis started. The physician in charge of the intensive care unit raised the suspicion of a viral haemorrhagic fever (VHF) such as the Ebola virus disease. A blood sample was taken and packed for dispatch to the reference laboratory in Conakry. The patient died at 10 p.m. with a picture of toxic-infectious shock and diffuse bleeding. The body was transported to the morgue and transferred to the Red Cross for a dignified and secure burial. 

### 3.2. Laboratory Diagnosis, Field Investigation and Contact Tracing

On 30 January 2020, the laboratory received the blood sample of the case and proceeded to various testing. Differential diagnosis included malaria, serology of HIV and YFV, and acute infection by HBV, EBOV and LASV. Only the presence of the LASV was confirmed on 30 January 2020 by conventional LASV RT-PCR. It was further re-tested by real-time LASV RT-PCR, and LASV Ct values of 36.0 for the S-segment and 24.5 for the L-segment were found. According to the equation of the standard curve for the GPC assay (y = 0.293x + 14.324 where y = log10 (RNA copies/mL plasma) and x = Ct), we can estimate the number of copies/mL to be 5.97E+03. The confirmation of a Lassa fever case launched field investigations and contact tracing activities. The investigation in Kissidougou revealed that the patient was an entrepreneur who was a timber trader. Before falling ill, he had spent a few days in Dandou, a village 32 km from Kissidougou, where he frequently went to do business. A total of 41 individuals in contact with the case in Kissidougou (*n* = 7), Mamou (*n* = 32) and Conakry (*n* = 2) were identified by the field teams, and further sampled for LASV RT-PCR as per Ministry of Health National Strategy guidelines in the case of VHF suspicion even in the absence of symptoms. None of them were found positive for EBOV or LASV by RT-PCR (Appendix A). 

### 3.3. Phylogeny

Phylogenetic analysis shows that the LASV strain from the case, further named the Kissidougou strain, belongs to clade IV of the *Lassa mammarenavirus* genus. This clade includes LASV known to circulate in Faranah (Upper Guinea) and in Macenta (Forest Guinea). Trees based on the combined complete glycoprotein (GP) and nucleoprotein (NP) (Figure 2A) and polymerase (Figure 2B) show that the Kissidougou strain is very closely related to a LASV strain identified in Liberia in 1981 (LIB-807987). Nucleotide similarities between these two strains are 88.9% for GP, 87.8% for NP and 86.7% for the polymerase. The amino acid translation gives similarities of 96.9% for GP, 96.0% for NP and 93.5% for the polymerase. The time of the most recent common ancestor (TMRCA) of the Kissidougou and LIB-807987 cluster is estimated at 139 (95% Highest Posterior Density (HDP) interval: 119–160) years using the GPC and NP, and at 155 (95% HPD interval: 139–170) years using the polymerase. In addition, the analysis on the partial NP, including four more sequences published in Bowen et al. 2000 [14], further supports that the Kissidougou cluster is different from that of Faranah, Macenta and N’zérékoré (Appendix A).

## 4. Discussion

### 4.1. Identification of the Kissidougou Sub-Lineage

In Guinea, numerous LASV sequences have been largely described in rodents, notably in the region of Faranah (221 sequences), Kindia (6 sequences) and Guékedou (9 sequences) [9,10,11,20,21]. However, only 5 sequences are derived from humans, 2 complete from Faranah and Macenta [22,23], and 3 partials from Kissidougou and Nzérékoré [14]. Our report represents thus the sixth description of a LASV strain isolated from humans in Guinea. It demonstrates the circulation of LASV in the surroundings of Kissidougou, which was already observed in 1960 among missionaries living in Telekoro near Kissidougou. Indeed, that historical study revealed that three individuals who had fever with long convalescence, with or without deafness, harboured LASV neutralizing antibodies [17]. The child of one of the missionaries also had LASV-neutralizing antibodies without signs of a febrile episode. From 1996 to 1999, an investigation among patients hospitalized in Kissidougou revealed that six of them were Lassa seropositive [12]. They came from villages around the town (Figure 1). This therefore indicates that LASV has been circulating for 60 years in the Kissidougou area. Phylogenetic analysis of the polymerase indicates an older origin, dating back 155 (95% HPD interval: 139–170) years. 

The strain described here is clearly related to the sequence LIB-807987 observed in 1981 in Liberia. It may be thus speculated that trades, including those related to the timber trade between Forest Guinea and Liberia, and commercial business favour LASV circulation between Kissidougou and Liberia. The case was, indeed, a logger merchant who used to sell wood in Forest Guinea without prior travel history to Liberia. Yet, a clear geographical origin (i.e., Liberia or Guinea-Kissidougou) for this sub-cluster remains speculative due to a lack of information about LIB-807987, as well as a lack of additional sequences. Thus, the name “Kissidougou cluster” is to highlight that it is different to that of the Macenta cluster but that it is also still part of the LASV known to circulate in Guinea. The Macenta cluster also includes three sequences, which, although identified in Liberia, originate from Macenta, Guinea, a town bordering Liberia. For example, the case described in 2018, the sequence of which is referenced under number LIB-LF18040 (MH215285 in Wiley et al. 2019), was a woman working at SOGUIPAH, near Macenta. She was diagnosed in Ganta, the neighbouring town in Liberia. The other large “Liberia” cluster, composed of sequences identified in, and originating from Lofa, Bong and Nimba counties in Liberia, forms a cluster independent from that of Guinea. Altogether, the description of this new strain allows us to speculate about a “Macenta” and a “Kissidougou” cluster circulating in Forest Guinea. These clusters appear to be more recent than the ones observed in Faranah, Upper Guinea, and in Liberia. This last country is probably the entry point of the virus in the Mano River region [24].

### 4.2. Strengths and Weaknesses of the Guinean Health System

In three weeks, the patient went through two health centres, one in Kissidougou and one in Mamou. Moreover, despite returning to the same health care facility in Kissidougou at a one week interval with worsened symptoms, no suspicion of VHF was considered. Even following his admission in a critical state at the CT-EPI of Mamou, the diagnosis of severe malaria continued to be considered. This indicates that the accurate and rapid clinical diagnosis of VHF in endemic areas is challenging, despite the 20142–016 Ebola virus disease epidemic in the country. This could be improved by providing regular awareness training on VHF case definitions and access to adequate communication tools (e.g., posters, flyers, drawings etc.).

On the other hand, the turnaround time for differential diagnosis was short, around two days (Appendix A). This included sample collection, transportation to the laboratories in Conakry and Guéckédou, analysis, results interpretation and reporting to the ANSS and WHO. Since the Ebola outbreak, the diagnostic chain has greatly improved and two days are now required to provide a diagnostic result of samples collected in remote areas. Similarly, field investigations and contact tracing activities, which have been in place since the creation of the ANSS in July 2014, four months after the start of the Ebola crisis in Guinea, have been rapidly launched following the alert and no secondary cases have been identified. Finally, the absence of secondary infections at the health centres visited by the case also indicates adequate infection prevention and control practices. 

## 5. Conclusions 

This case report shows that early identification of VHF cases remains challenging in remote areas. Regular awareness training may facilitate the implementation of improved field surveillance and early detection. The molecular description supports a new sub-lineage of LASV in Forest Guinea.

## Figures and Tables

**Figure 1 viruses-12-01062-f001:**
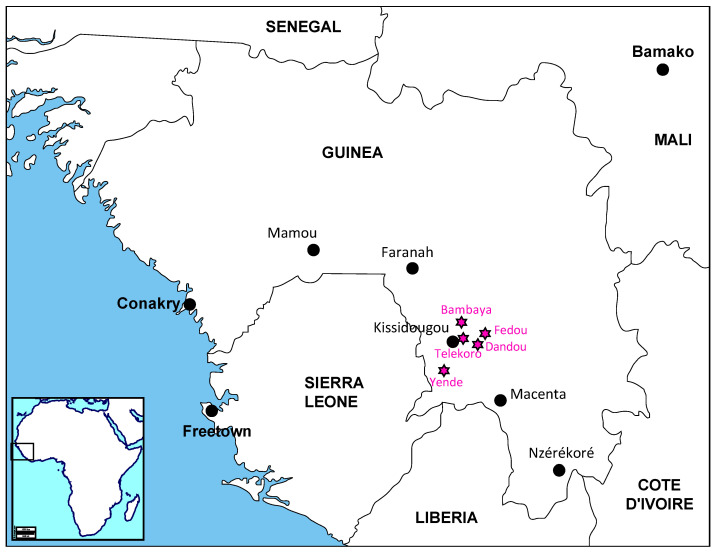
Map of Guinea showing the towns of interest, Kissidougou where the patient lived and Mamou where he died. Faranah, Macenta and Nzérékoré are the towns where Lassa virus sequences obtained from humans have been documented. The pink stars represent the locations of Lassa fever cases described in Henderson et al. 1972 [17], Bausch et al. 2001 [12] and this study.

**Figure 2 viruses-12-01062-f002:**
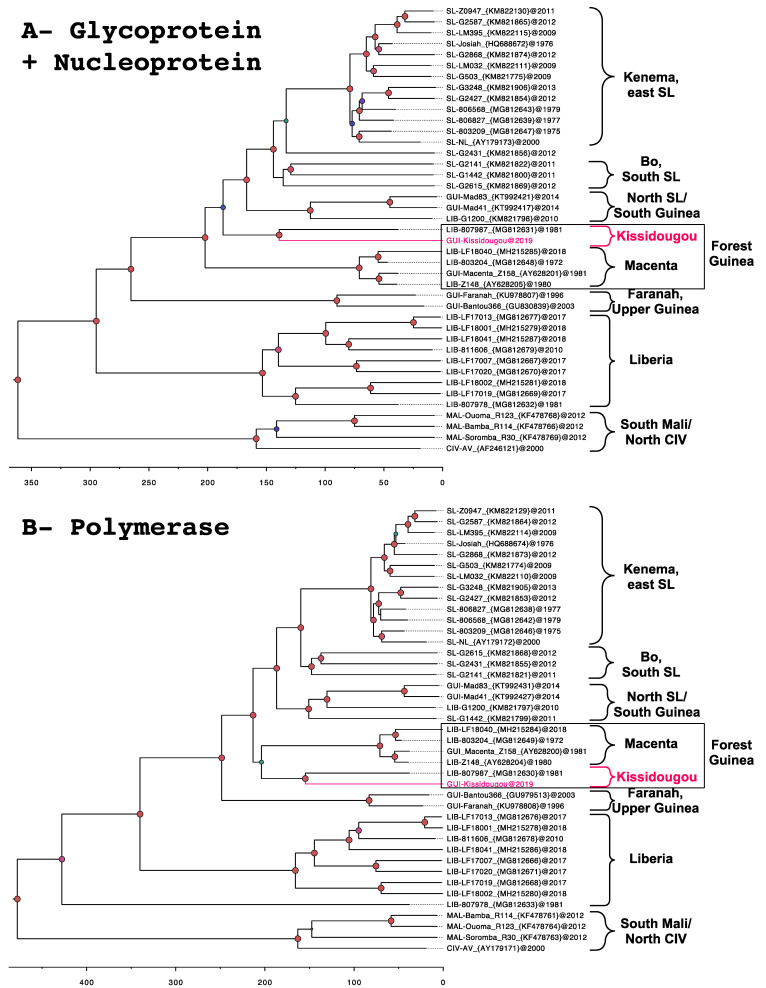
Time-calibrated phylogeny of the Lassa virus strains, gathering 41 sequences belonging to lineages IV and V (i.e., represented by “South Mali/North CIV”) and showing the new one from Kissidougou (in pink). The trees show the complete glycoprotein precursor and nucleoprotein in a linked model (**A**), and the complete polymerase (**B**). The trees were inferred by using the Bayesian Markov chain Monte Carlo method, with GTR + gamma model, strict clock, constant population size and partitioning into codon positions 1, 2, 3. Posterior probabilities are coded as follows: 0.7, green dots; 0.80 to < 0.85, purple-blue dots; > 0.97, red dots. Geographical origins are noted on the right of each tree, and scale axis represents the time in years.

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
