# Peer review of "A Sporadic and Lethal Lassa Fever Case in Forest Guinea, 2019"

_viruses, 2020, doi:10.3390/v12101062_

Round 1

Reviewer 1 Report

This study by Magassouba and colleagues provides valuable information about an acute case of Lassa fever convincingly diagnosed in Guinea.  The presence of RNA from the Lassa virus in patient serum is demonstrated by RT-PCR as well as next generation sequencing, and provides evidence of active Lassa infections in this area (as opposed to the presence of neutralizing antibodies, indicating only prior disease).  The description of the field investigation is helpful in this study, as is the discussion regarding the patient’s (multiple) initial diagnoses of malaria and it makes it clear that LASV is actively circulating and yet likely to be under-reported.  This case study highlights the need to consider a LASV diagnosis, and also the potential burden of Lassa fever that is unrecognized.

Major Points:

  • It would be helpful to show a standard curve for the LASV RT-PCR assay, to translate CTs of 36/24.5 into RNA copies/ml.

Minor Points:

  • Please specify if the presence of Hepatitis B virus or antibodies against Hepatitis B were tested using the rapid screening kit (line 70).
  • There is an additional “in” on 74.
  • The LASV test was presumably carried out with primer probes targeting the GP gene? (line 76). It would be good to clarify which tests are for antibodies and which are for genetic material in this section (as well as their gene targets).
  • I had difficulty accessing the complete sequences using the accession numbers provided (#MT861993, #MT861994).
  • Line 188, should read “used to sell wood”
  • The information in Table S1 appears to be duplicated inside the table (with two separate fields of view that can be scrolled through, but appear to contain duplicate information).

Author Response

This study by Magassouba and colleagues provides valuable information about an acute case of Lassa fever convincingly diagnosed in Guinea.  The presence of RNA from the Lassa virus in patient serum is demonstrated by RT-PCR as well as next generation sequencing, and provides evidence of active Lassa infections in this area (as opposed to the presence of neutralizing antibodies, indicating only prior disease).  The description of the field investigation is helpful in this study, as is the discussion regarding the patient’s (multiple) initial diagnoses of malaria and it makes it clear that LASV is actively circulating and yet likely to be under-reported.  This case study highlights the need to consider a LASV diagnosis, and also the potential burden of Lassa fever that is unrecognized. 

Major Points:

  • It would be helpful to show a standard curve for the LASV RT-PCR assay, to translate CTs of 36/24.5 into RNA copies/ml.

Response: We provided the translation of Ct value for the GPC assay according to the equation of the standard curve for the GPC assay (y = 0.293x + 14.324 where y = log10 (RNA copies/ml plasma) and x = Ct). This is now included into the text. We don’t have the equation of the standard curve for the L segment.

Minor Points:

  • Please specify if the presence of Hepatitis B virus or antibodies against Hepatitis B were tested using the rapid screening kit (line 70).

Response: this is now clarified in the section 2.2 (methods), in beginning the sentence by “the presence of antibodies …” or ” the presence of xx virus…”. We also modified the section 3.2 (results).

  • There is an additional “in” on 74.

Response: this has been corrected

  • The LASV test was presumably carried out with primer probes targeting the GP gene? (line 76). It would be good to clarify which tests are for antibodies and which are for genetic material in this section (as well as their gene targets).

Response: We now provided the details of the primers used in the conventional PCR. The protocol is published in Ölschläger et al. 2010, which is ref 17 in our article. We also clarified the tests done in serology and in PCR in both “Materials and Methods” and “Results” sections. See our response above.

  • I had difficulty accessing the complete sequences using the accession numbers provided (#MT861993, #MT861994).

Response: This is normal that you have difficulty to access the sequences because they will be released once the paper will be published.

  • Line 188, should read “used to sell wood”

Response: this has been corrected

  • The information in Table S1 appears to be duplicated inside the table (with two separate fields of view that can be scrolled through, but appear to contain duplicate information).

Response: The reviewer is right, the date are often duplicated to show that the test have been carried out the same day of the reception. Also, the shipment was often done the same day of blood collection. In the view to have a better understanding of the turnaround time, some of the remaining terms in French are now translated into English.   

Reviewer 2 Report

This case report by Magassouba et al documents a case of Lassa fever in Forest Guinea, a place not associated with endemic LASV. This well-written report explains the medical response to a patient showing up with symptoms that are unfortunately non-specific, especially early in the disease course. Only late infection were viral hemorrhagic fevers considered. Once considered the testing and tracing of contacts found no secondary cases. It was a very interesting read and the viral sequence suggested that the area may have some Lassa fever in the area that may go undetected because of lack of testing.

Author Response

We thank the reviewer for her/his positive review.